# Fluid Therapy in Acute Pancreatitis—Current Knowledge and Future Perspectives

**DOI:** 10.3390/ph18111601

**Published:** 2025-10-23

**Authors:** Miłosz Caban, Hubert Zatorski, Ewa Małecka-Wojciesko

**Affiliations:** Department of Digestive Tract Diseases, Faculty of Medicine, Medical University of Lodz, 90-153 Lodz, Poland; milosz.caban@stud.umed.lodz.pl (M.C.); hubert.zatorski@umed.lodz.pl (H.Z.)

**Keywords:** acute pancreatitis, fluid resuscitation, fluid therapy, normal saline, Ringer’s lactate

## Abstract

Acute pancreatitis (AP) is one of the most frequent diseases requiring hospitalization in gastroenterology or intensive care unit departments. Its incidence and hospitalization rates have steadily increased over the last few years, contributing to high costs of medical care. This disease is associated with relevant mortality and morbidity rates. Fluid therapy in the first 48–72 h has an important role in the clinical course and complications; however, it has been raising numerous controversies recently. We present a review article summarizing the current knowledge about fluid therapy in AP. The demonstrated results are based on the most recent clinical studies published in the last five years. Data confirms that the therapy should be individualized along with the amount of fluids adapted to body mass, concomitant diseases, critical signs, and laboratory markers. A relevant issue in the context of fluid therapy of AP is fluid resuscitation that should be implemented in some patients upon hospital admission to maintain organ perfusion and substrate delivery. Ringer’s lactate should be preferred in the vast majority of AP cases over normal saline solution. Its use is associated with lowered risk of intensive care unit admission and local complications development, reduced hospital stay, and decreased mortality. Colloids, mainly hydroxyethyl starch, should not be recommended. Moderate-rate fluid infusion seems to be an advantage over high-rate infusion. Relying on presented results, fluid therapy has a key therapeutic role in AP management.

## 1. Introduction

Acute pancreatitis (AP) is an acute inflammation of the pancreas and a leading disease of gastroenterological disorders. It belongs to the leading causes of hospital admission with approximately 300,000 emergency unit visits each year in the United States [1,2]. The most recent data demonstrate that the global incidence of AP is steadily growing [3]. It is estimated that the mean global incidence of AP is approximately 34/100,000 cases [4]. However, this rate is higher in some countries, including Poland, where incidence rate of AP is one of the highest (72.1/100,000 cases) [5]. Total AP mortality rate is estimated to be about 1% [4]. Nevertheless, in severe AP (SAP) with systemic inflammatory response syndrome (SIRS) and organ dysfunction (OD) or pancreatic necrosis, mortality may reach up to 40%, depending on the treatment quality [3,4].

Currently, recurrent attacks of AP remain a challenge and there are still many controversies regarding therapeutic modalities, such as pain therapy, time of feeding introduction, nutrition type, and antibiotic use. In particular, real-life management does not always correspond with the current guidelines, which were recently shown for the overuse of antibiotics and proton pump inhibitors (PPI) [6,7,8,9,10]. Nevertheless, fluid therapy is one of the most important cornerstones of AP management. It has a relevant, confirmed impact on the clinical course, prognosis, and potential complications of the disease [11]. The detailed strategies of AP therapy have changed over the years [12,13,14]. Nonetheless, the initial therapeutic management, including fluid therapy, in the first 48–72 h after diagnosis of AP is crucial [14]. It contributes to correcting loss of third-space volume and tissue hypoperfusion, as well as counteracting pancreatic and systemic microcirculatory impairment. Also, it was proven that adequate early fluid resuscitation seems to prevent pancreatic necrosis, OD, and SIRS, and preserve microcirculation of the pancreas [15]. So far, there are still doubts regarding the most beneficial type of fluids as well as the rate at which one should be administered. Normal saline (NS) and Ringer’s lactate (RL) are the two most commonly used fluids in AP. Although it has been a popular crystalloid for decades, NS does not seem to be a proper fluid, mainly due to the risk of metabolic acidosis aggravating the inflammatory response [16]. On the other hand, the rate of administered fluids is another controversy. Supporters of aggressive fluid therapy believe that it may reduce mortality. In contrast, other researchers argue that this type of therapy is associated with, among others, a high risk of fluid overload [17,18].

Hence, to gain a clearer understanding of the actual therapeutic management of fluid resuscitation and therapy in adult patients with AP, this review discusses updates on management strategies for AP based on the most recent scientific evidence. The aim of this study was a summary of actual knowledge of management of fluid therapy in AP, including the assessment of benefits and harms of different fluid therapy protocols, considering the types of fluids and rates of their administration. Also, we wanted to provide an actual stance on fluid management in AP. The study hypothesis was that fluid therapy should be individualized, and non-aggressive fluid therapy seems to have an advantage over aggressive fluid therapy, primarily in the context of the development of AP complications.

## 2. Search Strategy

PubMed, Google Scholar, Wiley, Springer, Scopus, Embase, and Web of Science databases were systematically and extensively searched for the bibliography. Clinical trials were also searched using the ClinicalTrials.gov database. The search included all the studies published up to August 2025, using the following keywords, alone or in combination: acute pancreatitis, severe acute pancreatitis, moderately severe acute pancreatitis, fluid therapy, fluid resuscitation, fluids, crystalloids, colloids, aggressive fluid therapy, non-aggressive fluid therapy, mortality, complications, safety, organ failure, SIRS, and pancreatic necrosis. The results of articles published in the last five years were crucial for creating this article. The searches were filtered to include only studies published in English. The bibliography was carefully selected based on the pertinence and relevance of the articles to use of fluid therapy in AP. The titles and the abstracts were independently screened by the investigators, and the selected papers were subsequently discussed with all the authors. Every article was carefully assessed, and various insights were gathered and studied to present an exhaustive perspective on the prevailing trends in fluid therapy in AP. Such a selection process ensures that this review is based on relevant research. In the last five years, most of the data has come from systematic reviews and meta-analyses. All eight available randomized, controlled clinical trials published in the last five years were included in this review article. Studies, including clinical trials and systematic review with meta-analyses, concerning fluid therapy in pediatrics and prophylactic treatment for post-endoscopic retrograde cholangiopancreatography were excluded. Nonetheless, we thoroughly examined each article, assessing the methodologies, results, and implications, which enabled us to summarize the current knowledge in the field, identify gaps, and propose directions for future research.

## 3. The Rationale for Fluid Therapy in AP

Local inflammation induced by intrapancreatic activation of proteolytic enzymes occurs in AP. The extent of the inflammatory response determines the severity of disease. The initial inciting event induces the injury of acinar cells. For example, in biliary AP, ductal obstruction with gallstone increases the intraductal pressure and blocks the usual acinar exocytosis of trypsinogen. It initiates a cascade of intra-acinar events, including zymogen activation, autophagy, oxidative stress, redox signaling disturbances, mitochondrial dysfunction, and endoplasmic stress, which lead to the premature activation of intracellular pancreatic enzymes [19]. The zymogen activation with stimulation of calcium signaling and co-localization of lysosomes are main mechanisms in gallstone-related and alcoholic AP [20]. The injury of acinar cells and their necrosis are associated with the release of damage-associated molecular pattern (DAMP) being self-antigens, such as dsDNA, ATP, high mobility group box 1 protein (HMGB1). They activate specific receptors, known as pattern recognition receptors (PRRs), localized on the surface of innate immune cells. Under the influence of DAMP, the activated cells release cytokines and induce sterile inflammation [19]. Subsequently, the up-regulation of vascular adhesion molecules occurs, facilitating the infiltration of pancreatic parenchyma by leukocytes, which initiates systemic inflammation [21]. The balance between the anti-inflammatory and pro-inflammatory mediators has a key role in the development of potential systemic complications, such as circulatory disturbances, secondary infections or SIRS, single OD syndrome (SODS), or multiple OD syndrome (MODS) [22]. In addition, microcirculatory alterations, coagulation disturbances, and subsequent cellular and tissue damage contribute to organ dysfunction. The microcirculatory abnormalities result from primarily leukocyte adhesion, platelet aggregation, endothelium dysfunction, and hemodynamic changes [23,24,25]. Those disturbances may lead to decreased pancreatic tissue perfusion and, consequently, increased capillary permeability. In consequence, vessel damage and extravascular fluid accumulation, pancreatic edema, ascites, or pleural effusion usually lead to hypovolemia and systemic shock. Therefore, bowel ischemia may occur followed by gut barrier dysfunction, enhanced intestinal permeability, and bacteria translocation with secondary infections of pancreatic parenchyma and other organs. Accompanying the symptoms, mainly vomiting and abdominal pain, oral feeding and oral intake of fluids are critically limited, enhancing fluid depletion. In addition, tachypnoea and diaphoresis related to pain, fever, or SIRS augment the fluid loss [19,26,27]. These changes have a vicious character cycle and exacerbate fluid loss [28]. On the other hand, the accumulation of fluid in the peritoneal cavity and retroperitoneal space may limit bowel movements and cause the mechanical compression of the intestine, fluid accumulation in the intestinal lumen, and even abdominal compartment syndrome or ileus [29,30]. Therefore, the amount of administered fluid must be carefully and precisely evaluated (Figure 1).

## 4. Severity of AP

The determination of AP severity is important in fluid management. In addition, the prognosis of AP severity at hospital admission is key for prompt treatment, close monitoring of severe patients, choice of fluid rate, and the volume and decision regarding the implementation of fluid resuscitation [2]. Therefore, the administration of fluid must be adjusted to the evaluation of AP severity to improve organ functions and to save from death. There are some clinical features and parameters at admission that are associated with an increased initial risk of severe disease course. Also, several AP-specific severity scoring systems for risk stratification and prediction of SAP have been widely developed for many years [31] (Table 1). It is worth emphasizing that it cannot be used serially to successfully indicate the AP severity and therefore determine the choice of fluid resuscitation. These scores should not supersede clinical assessment. Their utility rather lies in excluding SAP. Nonetheless, the Acute Physiology and Chronic Health Evaluation (APACHE II), the Ranson criteria, the Bedside Index of Severity in AP (BISAP), the Simplified Acute Physiology Score (SAPS II), Sequential Organ Failure Assessment (SOFA), CT severity index (CTSI), the Glasgow (Imrie) score, and harmless AP score (HAPS) are the most recommended [32]. Some of them require imaging for evaluation. For example, CTSI determines severity of AP and the risk of death based on CT imaging that is not routinely performed at the admission. Briefly, the APACHE II is a severity-of-disease classification system, one of several intensive care unit (ICU) scoring systems. The classification is based on the results of 12 routinely performed measurements of vital signs and physiological parameters, among others such as heart rate, respiratory rate, body temperature, mean arterial pressure, arterial blood pH, and serum level of creatinine and electrolytes. In turn, the Ranson criteria form a clinical prediction rule for predicting the prognosis and mortality risk of AP. The BISAP predicts the severity of AP at the hospital admission, and a score of three or more indicates SAP. In contrast, Glasgow (Imrie) score is a modification of the Ranson criteria and determines the severity of AP based on age and seven laboratory values. Three or more points correspond to the risk of SAP [31]. Although a lot of scoring systems are available, their sensitivity, specificity, and utility for predicting SAP, OD, ICU admission, and complications including pancreatic necrosis differ. So far, several studies, mainly retrospectives, have been conducted to compare their prediction efficacy [33,34,35,36,37]. Based on these results, APACHE II (the cut-off value ≥ 8) and Ranson (the cut-off value ≥ 3) had the highest accuracy in predicting SAP, mortality, MODS, and ICU admission, especially in younger patients with AP [33,34]. The SOFA (the cut-off value ≥ 7) was generally characterized by the lowest sensitivity for predicting SAP [37]. BISAP (the cut-off value ≥ 2) offered commendable accuracy and applicability in predicting severity and local complications. However, it was not a good scoring for mortality prediction [34,37]. It seems that the scoring APACHE II, Ranson, and BISAP scales are good scales for selection of patients with SAP. Consequently, they could facilitate an adequate choice of fluid administration.

## 5. General Characteristics of Fluid Management in AP

Early administration of intravenous fluids immediately after hospital admission is necessary to correct the loss of third-space volume and tissue hypoperfusion. It is essential to emphasize that detailed, comprehensive anamnesis and a careful physical examination are obligatory on admission. In most of the cases, AP occurs in male patients regularly consuming large amounts of alcoholic beverages. These patients tend to be more resistant to abdominal pain and enter the emergency unit only several days after the symptoms begin. It may falsify a credible, clinical assessment that these patients are extremely dehydrated and hypovolemic. In the context of initial fluid management, adequate assessment of the presence or lack of hypovolemia is key for further medical treatment. The signs detected during physical examination, mainly hypotension, tachycardia, dryness of the mucosae, extended capillary refill, skin mottling score, and oliguria, indicate hypovolemia. Also, results of laboratory tests, including elevated levels of HCT, BUN, creatinine, and lactates, may confirm hemoconcentration and subsequently the hypovolemic status of patients. The confirmation of hypovolemia is a direct indication for starting fluid resuscitation [38]. Fluid resuscitation is a special form of fluid therapy. It is used in life-threatening situations where rapid fluid replenishment is necessary to stabilize the patient’s condition [39]. The fluid resuscitation in AP should be immediately started with a bolus of 5–20 mL/kg over 30–60 min [38]. The significant issue remains with the type of fluid used in fluid resuscitation. Evidence suggests that the use of colloids, including gelatin or hydroxyethyl starch (HES), in fluid resuscitation is associated with increased risk of morality and acute kidney injury (AKI), coagulopathy, and pulmonary edema compared to crystalloids, limiting their use. In contrast, albumin, a type of colloid, is characterized by comparable safety with crystalloids, though it has no clear superiority in terms of resuscitation efficacy. Also, its use is related to high costs [39,40,41]; that is why crystalloids are preferred. After administration of the first bolus, a reassessment should be conducted. Hemodynamic stabilization and rapid optimization of organ function are the initial goals of resuscitation. In the case of persistent hypovolemia, the next bolus of fluid should be administered, and colloids, primarily gelatin, should be considered due to the longer stay in the intravascular compartment and effective increase in oncotic pressure. Colloid use should be especially implemented in the case of HCT < 25% and severe hypoalbuminemia (<2 g/dL) [41]. The persistent, refractory hypovolemia after fluid boluses is an indication for ICU consult and the subsequent transfer to the ICU for vasopressor and advanced hemodynamic monitoring to maintain MAP ≥ 65 mmHg. In addition, additional fluid boluses (5 mL/kg over 30 min) as needed after thorough assessment of circulatory status should be considered, especially in those with lactate > 4 mmol/L and skin mottling [38].

After fluid resuscitation (if was needed), adequate fluid therapy should be carried out. There are two main types of fluid therapy in AP: aggressive (3 mL/kg/h) and conservative, known as non-aggressive (1.5 mL/kg/h). Generally, aggressive fluid therapy seems to be beneficial in subjects with predicted mild AP (MAP) when started within the first 4 h from the hospital admission and continued for 12–24 h. The delay of aggressive intravenous fluids administration in MAP beyond the therapeutic window of 4–6 h may be harmful and increase mortality rate or risk of the development of serious pancreatic and systemic complications, including pancreatic necrosis, MODS, and ARDS [15]. Early aggressive intravenous fluid therapy during the first 24 h of MAP is associated with an improvement in survival and minimizes occurrence of pancreatic necrosis [42]. Nonetheless, fluid therapy in AP is, currently, goal-directed therapy based on the maintenance of appropriate cardiovascular, renal, and pulmonary function, as well as electrolyte balance and tissue perfusion. It has been a structured approach in directing fluid administration by specific physiological targets. The administration of intravenous fluids should be guided by clinical and biochemical parameters such as the heart rate, mean arterial pressure, central venous pressure, urine output, central venous oxygen saturation, blood urea nitrogen concentration, and hematocrit and lactate levels [43,44]. The level of lactate reflects the tissue hypoperfusion, thromboembolic risk, and consequently indicates on the risk of local and systemic complications [43,45,46,47]. It is worth emphasizing that the patients with MAP or MSAP are treated in gastroenterological departments. However, SAP, especially with hypovolemia, should be hospitalized in ICU. The gastroenterological staff does not perform close monitoring of the clinical parameter and cannot adapt the fluid administration to these changes. It results from the lack of specialized equipment. In contrast, the ICU staff takes care of many different kinds of medical and surgical patients. In addition, it needs more standardized and less individualized treatment management. That is why the ability to achieve and inspect the goals of fluid therapy differs between these two medical departments. Neutral fluid balance, normalization of HCT, BUN, lactates, and prevention of hypoperfusion and complications are goals that could be achieved in the gastroenterological departments. Due to the fact that the patients with SAP are constantly monitored in ICU, MAP ≥ 65 mmHg, urine output > 0.5 mL/kg/h, capillary refill time < 3 s, the lactate level < 2 mmol/L, HR < 120/min, and central venous oxygen saturation (ScvO_2_) ≥ 70% are aims of fluid therapy in ICU [32,38].

In the patients with MODS, fluid therapy should be particularly careful and individualized to avoid under- or over-treatment due to higher risk of mortality compared to those without MODS [48]. Maintaining balance between fluid therapy and complication occurrence is a challenge. It seems that local complications, such as acute fluid collection and pancreatic necrosis, are more frequent than systemic complications. The respiratory insufficiency seems to be the most frequent systemic complication, followed by renal insufficiency and circulatory failure [49]. Currently, general strategy with 1.5 mL/kg/h of RL infusion in the first 24–48 h, preceded by bolus of 10–20 mL/kg within first 2 h, is recommended in the occurrence of at least one of the following factors: prediction of moderately severe SAP based on prognostic scales such as APACHE II, Ranson score, or BISAP, age < 40 years old, alcoholic etiology, HCT ≥ 44% or BUN > 20 mg/dL, and signs of hypovolemia and AKI at the admission [50]. However, it must be emphasized that this strategy should also be individually adjusted and modified depending on patients’ clinical history. Elderly patients aged > 65 years and those with a history of liver cirrhosis, cardiac and/or renal diseases, including myocardial infarction, heart failure, and chronic kidney disease, require restriction of fluid therapy. It was demonstrated that aggressive, liberal fluid intravenous resuscitation of more than 15 mL/kg in the first day in patients with severe settings and a history of heart failure increases the risk of mortality. It probably results from excessive heart overload, significant impairment of cardiac contractility, or low ejection fraction [51]. The subjects with AP and heart failure or a history of myocardial infarction impairing heart function require 1000–1500 mL of fluid with or without an initial bolus of 10 mL/kg at hospital admission. Also, AP concomitant with oliguric or anuric AKI requires a reduced infusion rate of fluids. Fluid balance disturbances may lead to high morbidity and mortality due to potential pulmonary edema, hypoxia, and cardiac dysfunction. In addition, in these patients, the application of loop diuretics may be necessary [52]. In these subjects, detailed clinical and laboratory assessment should be performed after 24 h and excessive, aggressive fluid therapy beyond that time should be avoided. Taking the above considerations into account, hypertension, cardiogenic shock, congestive heart failure, anuric AKI, increased intracranial pressure, specific electrolytes imbalances, pulmonary edema, and some types of trauma are main contraindications to aggressive fluid therapy.

Furthermore, the etiology of AP may determine the volume and type of used fluids. The AP induced by hypertriglyceridemia or IgG4-related disease AP needs specific therapies and fluid therapy should not be aggressive due to the high risk of potential complications. On the other hand, RL is not a basic fluid in AP with ketoacidosis (usually as a decompensation of diabetes mellitus). In turn, hypercalcemia-induced AP is a contraindication for the use of RL due to the high content of calcium (3 mEq/L of calcium). In these cases, NS solution is recommended [53].

The stopping fluid therapy remains a challenge for clinicians. In addition, endpoints of fluid therapy are dependent on some factors, including the patient’s status. The occurrence of fluid overload is an indication to stop or decrease the administration of fluids, and this decision should be confirmed by a neutral fluid balance. In turn, the correction of dehydration, achievement of patient’s clinical stability, and tolerance of oral feeding for more than eight hours may lead to discontinuation of fluid therapy [32,38].

## 6. Fluid Rate and Volume

In the past, it was suggested that aggressive intravenous hydration at 5–10 mL/kg/h for 30–60 min (in severe volume depletion of even 20 mL/kg/h) and its continuation at 3 mL/kg/h for up to 48 h should be beneficial in all types of AP [42]. The randomized trial evaluated the initial management of patients with MAP, without SIRS or MODS. The aggressive hydration (20 mL/kg bolus followed by 3 mL/kg/h) with RL caused clinical improvement within 36 h, defined as the combination of decreased mean hematocrit, BUN, and creatinine, and reduced pain and improved tolerance of oral diet. In addition, clinical improvement was observed in a higher number of patients compared to the non-aggressive hydration (10 mL/kg bolus followed by 1.5 mg/kg/h) at 36 h from the disease onset (70% vs. 42%; *p* = 0.03). Also, persistent SIRS occurred less frequently in subjects with aggressive fluid therapy compared to standard fluid therapy (7.4% vs. 21.1%) [54]. In addition, a retrospective cohort study performed by Doshi et al. showed that early aggressive hydration contributed to lower opioid dose on the last day of hospitalization compared to standard hydration (23.9% vs. 35.3%; *p* = 0.044) in patients with MAP. The potential mechanisms of this phenomenon were the improvement in perfusion, increase in pancreatic and splanchnic blood flow and maintenance of hemodynamic balance. Also, the AP patients after aggressive hydration were less likely to be readmitted for any reason within 30 days [55]. Interestingly, a meta-analysis of five randomized control trials from 2022 showed that the aggressive intravenous hydration protocol using RL solution in MAP seemed to be more effective in reducing the risk of SIRS or MODS, epigastric abdominal pain, and the length of hospital stay (*p* < 0.001) compared to standard hydration. Nonetheless, this meta-analysis comprised the small number of studies (five randomized controlled trials) [56]. On the other hand, a randomized controlled trial from 2020 year compared “non-aggressive” versus an “aggressive” hydration in MAP, moderately severe AP (MSAP) and SAP with Hartmann’s solution. No differences in the frequency of persistent SIRS (*p* = 0.528), pancreatic necrosis (*p* = 0.710), respiratory complications (*p* = 0.999), AKI (*p* = 0.714) and prolonged hospital stay (*p* = 0.892) between the two groups were observed [57].

However, actually, the most recent studies, including systematic reviews and meta-analyses, suggest that aggressive fluid therapy may increase the risk of MODS or respiratory complications. Moreover, aggressive fluid therapy in SAP may result in increased mortality [44,58,59].

The multicenter, open-label, parallel-group, randomized, controlled trial (NCT04381169) from 2022 proved that early aggressive fluid resuscitation using RL solution was associated with increased incidence of fluid overload without improvement in clinical outcomes. In this study, a total of 249 patients with AP derived from 18 centers were randomly assigned in a 1:1 ratio to two groups: aggressive fluid resuscitation (a bolus of 20 mL per kilogram of body weight, followed by 3 mL per kilogram per hour) and moderate fluid resuscitation (a bolus of 10 mL per kilogram in patients with hypovolemia or no bolus in patients with normovolemia, followed by 1.5 mL per kilogram per hour in all patients). All subjects were assessed five times: at 3 h as an initial physical assessment evaluating fluid overload and at 12, 24, 48, and 72 h as biochemical and physical assessment. Fluid resuscitation was modified on an ongoing basis considering the patient’s clinical status. The fluid resuscitation was stopped after 8 h tolerance of oral feeding (in aggressive treatment group usually 48 h after randomization and in moderate treatment group usually 20 h after randomization). Results demonstrated no statistical differences in the frequency of MSAP and SAP, as well as MODS occurrence between the two groups. The fluid overload, defined as the presence of at least two of the three criteria (heart failure symptoms, hemodynamic or imaging evidence of heart failure), was observed in 20.5% of the subjects receiving aggressive resuscitation and in 6.3% of those receiving moderate resuscitation (*p* = 0.004). Moreover, the median duration of hospitalization was longer in the aggressive resuscitation group compared to the moderate resuscitation group (6 vs. 5 days) [59]. Other data confirm that aggressive intravenous fluid therapy in SAP increased the mortality risk as well as fluid load-related complication risk in both SAP and MAP. This therapy may substantially increase the risk of fluid overload, pancreatic necrosis, mortality, MODS, infections, AKI, pulmonary edema, respiratory failure, admission to ICU, and longer hospital stay [60,61,62,63,64,65].

It seems that aggressive hydration contributes to the increase in the vascular leak causing fluid sequestration, tissue hypoxia, and further congestion. All those may lead to OD and pancreatic ischemia, and even necrosis. The mechanisms contributing to AKI accompanying the aggressive fluid resuscitation in AP are excessive renal congestion, intravascular fluid accumulation, and decreased renal perfusion due to visceral edema and renal vascular congestion. In addition, the use of diuretics in fluid overload may additionally cause the AKI development [60,61,62,63,64,65]. Kumari et al. in 2024 assessed the optimal, early fluid therapy regimen in the patients with AP of different severity [66]. The high, moderate, and low fluid therapies were defined as ≥20 mL/kg/h, ≥10 to <20 mL/kg/h and 5 to <10 mL/kg/h, respectively. The results revealed improved clinical outcomes, such as bolstered tissue perfusion, and reduced hospital stay with low compared to moderate fluid therapy [66]. Most recent studies confirm the superiority of controlled, non-aggressive fluid therapy to aggressive fluid therapy with respect to both efficacy and safety outcomes. The aggressive intravenous hydration may significantly improve clinical outcomes during the first 24 h in the patients with MAP. Angsubhakorn et al. performed randomized clinical trial, in which 44 patients with MAP were divided into two groups: aggressive (20 mL/kg bolus followed by 3 mL/kg/h) and standard (10 mL/ kg bolus followed by 1.5 mL/kg/h) intravenous hydration with RL. In obese patients, clinical improvement at 24 h, defined as decrease in hematocrit, BUN and serum creatinine from baseline, decrease in epigastric pain level, and tolerance of oral intake, was noted in 72% of patients in the aggressive fluid group compared to 23% in the standard fluid group (*p* = 0.015) [67].

In the last five years, one clinical trial was published in the context of fluid resuscitation via colon in AP. Ni et al. analyzed two groups with SAP: patients with intravenous fluid therapy and, additionally, fluid resuscitation via colon. The fluid resuscitation via colon was administered by a disposable colonic enema tube inserted through the anus to a depth of approximately 25 cm and connected to an infusion device. Colonic liquid infusion using RL, NS, or pure water was applied at a rate of 250–500 mL per hour with fluid intravenous resuscitation at the same time until the goal of blood volume expansion was achieved. This goal was defined as the fulfillment of two or more of the following requirements: heart rate  <  120 beat per minute, mean arterial pressure 65 to 85 mmHg, urine output  >  0.5 mL/kg/h, hematocrit of 30% to 35%. The inflammation markers, such as CRP, procalcitonin, and white blood count, were significantly lowered in the group with colonic resuscitation than the remaining one. In addition, in the colonic treatment group, the frequency of mechanical ventilation and hypernatremia were notably decreased. However, there was no difference in the 90-day survival between both groups. These results suggest that intravenous fluid resuscitation supported by colonic resuscitation may exert beneficial effects, however, prognosis of patients seems to be similar [68]. In 2023, Liu et al. published a systematic review and meta-analysis assessing the efficacy and safety of early intravenous and enteral fluid resuscitation in SAP. The results confirmed that the enteral route of fluid administration in SAP reduced the incidence of new OF (*p* < 0.00001), persistent OF (*p* = 0.0003), mechanical ventilation (*p* = 0.01), ICU (*p* = 0.02), and pancreatic infection (*p* = 0.02) compared to intravenous only fluid administration. Nonetheless, the rate of mortality and surgical intervention did not differ between these two groups [69].

Oblivious to the above-described data, non-aggressive fluid therapy remains a standard approach, especially in SAP, but it must be administered under the actual clinical status of the patients with AP and their medical history. The main aims of fluid therapy are not allowing the BUN and hematocrit to rise within the first 24–48 h and to not develop SIRS and organ insufficiency. Fluid rate and volume should also be dependent on AP etiology. Further studies are necessary to confirm the utility of intestinal and colonic fluid resuscitation for the improvement in AP management.

The selected most recent studies assessing the rates and volumes of fluid therapy in AP are summarized in Table 2.

## 7. Fluid Type

Different fluids are being considered in hydration in AP. Colloids, including human albumin solution, HES, gelatine and dextran solutions, are not recommended due to possible adverse effects like AKI, pruritis or skin rash, and no proven survival benefits [72,73]. Therefore, crystalloids, both isotonic as represented mainly by NS and balanced/buffered, such as RL solution or Plasma-Lyte, are preferred. The adverse effects of NS administration, mainly hyperchloremic non-anion gap acidosis, became the main limitation for its use. Recently, the intravenous administration of RL in AP was associated with lowered risk of ICU admission, mortality, and local complications development and reduced hospital stay [74,75,76,77,78,79,80]. In addition, the use of RL for initial resuscitation in AP, compared to NS, lowered the 1-year mortality after disease onset [81].

There are few advantages of RL over NS in AP management. RL solution contains sodium, chloride, potassium, calcium and lactate in the form of sodium lactate, mixed into a solution with an osmolarity of 273 mOsm/L and pH of about 6.5. In contrast, the osmolarity of NS is approximately 286 mOsm/L and pH of about 5.5. Its administration is not only useful for increasing intravascular volume, preload, and perfusion, but also to prevent the development of non-anion gap hyperchloremic metabolic acidosis. Also, it provides sodium lactate as a bioenergetic fuel in ischemic conditions, reducing cell death. The greater pH of RL than NS may induce metabolic alkalosis and prevent premature trypsinogen activation that requires a low pH and determine the development of AP [82]. In addition, RL solution may be safely administered in septic shock accompanying AP without the risk of worsening lactic acidosis. It results from the fact that RL solution does not contain lactic acid, but sodium lactate [83]. Moreover, the calcium contained in RL binds ionically with non-esterified fatty acids that are associated with severity of the disease, and their level is often increased in the disease. Consequently, calcium is able to cause their saponification and reduce lipotoxic necrosis [84]. Also, it was presented that the infusion of large volumes of NS can cause impairment of vascular and renal function, as well as exacerbate abdominal discomfort in various settings, including healthy volunteers. It suggests that NS may exacerbate the abdominal pain in AP [85]. In addition, infusion of NS altered respiratory function in healthy subjects by an increase in small airways resistance and interstitial pulmonary edema measured by lung ultrasound [86,87]. Thus, NS may elevate the risk of occurring acute respiratory distress syndrome in AP. Data indicate that RL may exert an anti-inflammatory effect in AP. It was suggested that RL may affect the mitigation of systemic inflammation by a pH-mediated effect based on the lesser reduction in serum bicarbonate compared to the NS [88]. Furthermore, RL may suppress inflammation in AP by exerting a protective immunologic effect through the regulation of Toll-like receptor (TLR)-mediated inflammatory response [89,90].

In recent years, a lot of studies were conducted to verify and evaluate adequate choice of optimal fluid for intravenous hydration in AP. In 2018, meta-analysis by Iqbal et al. revealed that RL solution had anti-inflammatory activity in AP and advantage over NS, decreasing the SIRS risk. Nevertheless, the authors did not show a decline in mortality with RL compared to NS [91]. On the other hand, other meta-analyses did not show differences in SIRS occurrence in RL group compared to NS [74,92,93,94].

The most recent multicenter, stepped-wedge, cluster-randomized trial published in 2025 demonstrated the advantage of balanced multielectrolyte solutions over NS. In the group of 259 patients with predicted SAP, it has been showed that the patients who received Sterofundin^®^ (B. Braun, Melsungen, Germany) for initial fluid therapy were characterized by less frequent SIRS (17.0% vs. 29.3%, *p* = 0.024), longer period of MODS-free days (3.9 vs. 3.5 days, *p* = 0.00024) and ICU-free days (26.4 vs. 25.0 days, *p* = 0.0092) and stayed outside of the hospital more (19.8 vs. 16.3 days *p* < 0.0001) by trial day 30 compared those in NS group. In addition, Sterofundin fluid therapy was associated with the reduced mean serum chloride level on trial day 3, indirectly decreasing the risk of hyperchloremic acidosis [95]. Currently, there are no absolute contraindications to RL. Nonetheless, special care should be taken in patients with liver dysfunction that may predispose to accumulation of lactate. Also, RL as isotonic solution should not be applied in the subjects with cerebral edema that may be exacerbated with this therapy [83]. Data show that the toxicity of RL solution is primarily associated with fluid overload related to inadequate intravenous volume administration that may lead to consequences, from mild peripheral edema to respiratory distress secondary pulmonary edema [83]. In 2026, the WATERLAND clinical trial (NCT05781243) is expected to provide final evidence on which type of fluid should be ideally administered in AP [96].

So far, the benefits of RL have been quantified. The results derived from the two biggest, most recent meta-analyses deserve special attention and comparison. Dawson et al. conducted meta-analysis using a random-effects model. It involves twenty studies comprising 3752 AP patients. In the context of comparison of RL and NS, five randomized clinical trials and two single-center retrospective studies with 651 subjects were assessed on the type of fluid. In detail, 359 were treated with NS, 251 with RL, and 21 with HES. The results demonstrated that there was no difference in mortality between NS and RL (3.17% vs. 3.01%; *p* = 0.23), though patients treated with NS had a significantly longer hospital stay lengths (*p* = 0.009) [77]. In contrast, a second meta-analysis of nine studies was conducted by Hong and co-researchers comprising 1424 patients with AP in LR (*n* = 651) and NS (*n* = 773). The results showed that subjects treated with RL were less likely to develop MSAP/SAP (OR = 0.48, 95%Cl 0.34 to 0.67; *p* < 0.001), as well as had fewer local complications (OR = 0.54, 95%Cl 0.32 to 0.92; *p* = 0.02) and shorter hospital stay lengths (−1.09 days; 95%Cl −1.72 to −0.47 days; *p* < 0.001) compared to those treated with NS [80]. By comparing the results of both meta-analyses, six studies included were common, and the results of the second article were more extensive. Also, their limitations are similar. Not all the included studies were randomized clinical trials. The included studies did not contain individual, detailed information about the etiology and severity of AP. It may contribute to a high risk of misclassification bias wherein causation cannot be established with certainty. In addition, the credibility of data interpretation may also be hindered by the lack of uniform standardized fluid administration regimen across studies (mainly fluid rate and volume, the delay in starting resuscitation from the onset of AP) [77,80]. However, their results strongly suggest the advantage of RL over NS in fluid therapy in AP.

The selected most recent studies investigating the individual types of fluids administered in AP are summarized in Table 3. Oblivious to the above data, RL seems to be the preferred fluid in AP, and its use is associated with a lower risk of complications compared to NS.

## 8. Conclusions

The adequate initial treatment of AP has a key significance for the clinical course of the disease. Fluid therapy with proper pain control is the main strategy for the AP treatment. The fluid therapy should be individualized and based on actual fluid intake needs, body weight, obese status, comorbidities, age, and detailed clinical and laboratory assessment at the hospital admission (Figure 2). However, it is worth emphasizing that goal-directed fluid therapy is difficult and remains a challenge. It mainly results from the fact that individualized fluid therapy requires highly qualified medical personnel and close monitoring of the clinical parameters. Consequently, it is practically possible in only the ICU, where only SAP is treated. The goal-directed fluid therapy for patients with MSAP and MAP seems to be difficult to implement and conduct due to their hospitalization in gastroenterological units. On the other hand, fluid resuscitation may have a key significance for prognosis and mortality of AP. It should be applied in all patients who require it after detailed anamnesis and of a careful physical examination at the hospital admission. Unfortunately, inadequate clinical assessment is relatively common. The choice of appropriate type and ratio of fluids also has an important role for AP. RL solution is regarded as the dominant fluid in fluid therapy used in the AP. It may prevent local complications and reduce mortality and hospital stay compared to NS. Nonetheless, RL is not dedicated for all patients with AP, especially hypercalcemia-induced AP. Further well-designed randomized, clinical trials, including those comparing various type of fluids, are necessary to improve actual treatment results. An interesting open-label multicenter randomized controlled trial: the WATERLAND trial study protocol (NCT05781243) is in progress, and its results are in preparation. This review has a narrative character and some limitations. It primarily adopts a narrative synthesis approach without conducting quantitative heterogeneity assessment or pooled analysis of effect sizes. The significant heterogeneity among studies and the absence of quantitative synthesis is a major limitation of this article. Consequently, the robustness of its conclusions may be somewhat weakened.

## Figures and Tables

**Figure 1 pharmaceuticals-18-01601-f001:**
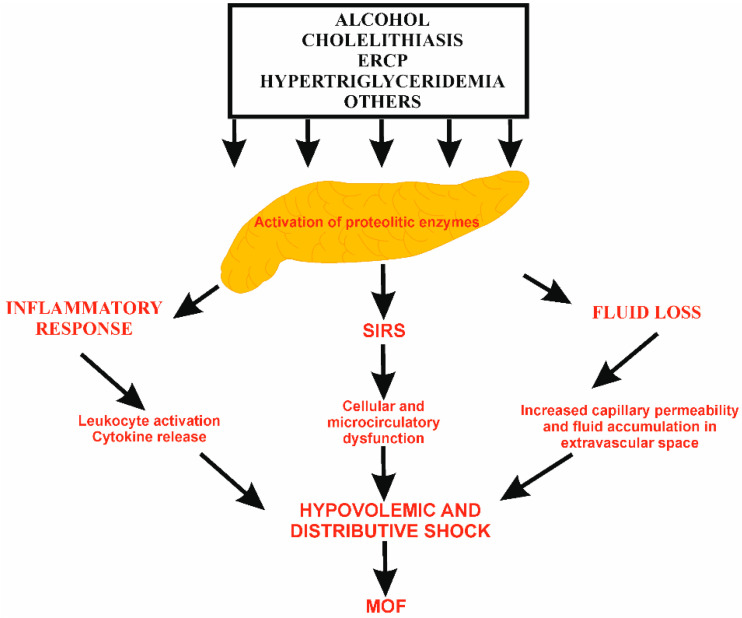
The pathophysiology of fluid loss in AP. The intrapancreatic activation of proteolytic enzymes leads to local pancreatic inflammatory response with leukocyte activation, cytokine release, and oxidative stress. The massive release of pro-inflammatory factors causes systemic inflammation, SIRS, increased capillary permeability, and fluid accumulation in extravascular spaces. ERCP, endoscopic retrograde cholangiopancreatography; MOF, multiple organ failure; SIRS, systemic inflammatory response syndrome.

**Figure 2 pharmaceuticals-18-01601-f002:**
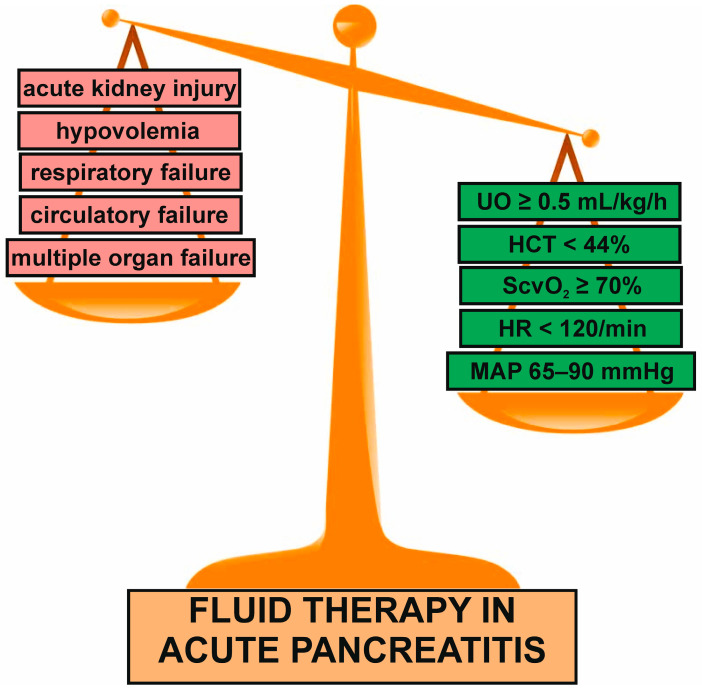
Schematic view of targets and potential threats of fluid therapy in acute pancreatitis. Fluid therapy should be adequately conducted to achieve targets and avoid complications.; HCT, hematocrit; HR, heart rate; MAP, mean arterial pressure; ScvO_2,_ central venous oxygen saturation; UO, urine output.

**Table 1 pharmaceuticals-18-01601-t001:** Clinical features and parameters on admission associated with increased risk of severe disease.

Clinical Features and Parameters of Adverse Prognosis in AP	
Characteristic of patient	Age > 55 years
	BMI ≥ 25 kg/m^2^
	Alcohol abuse
	Altered mental status
	Comorbid diseases
SIRS (≥2 criteria)	Body temperature < 36 °C or > 38 °C
	HR > 90/min
	RR > 20/min (PaCO_2_ < 32 mmHg)
	WBC > 12 G/L or < 4 G/L (or > 10% immature leukocytes)
Laboratory tests	HCT ≥ 40% (women)/≥ 44% (men)
	Glucose > 200 mg/dL
	Calcium < 1.97 mmol/L
	CRP > 15 mg/dL
	BUN ≥ 20 mg/dL
	LDH > 350 U/L
	Increased level of creatinine
Radiological imaging	Pleural effusions
	Pulmonary infiltrates
	Multiple or extensive peripancreatic fluid collections
Scoring scales/systems	BISAP ≥ 3 points
	APACHE-II ≥ 8 points on admission or within first 72 h
	SOFA elevation ≥ 2 points
	Ranson criteria ≥ 3; assessment on admission and after first 48 h

AP, acute pancreatitis; APACHE-II, Acute Physiology and Chronic Health Evaluation II; BISAP, Bedside Index of Severity in Acute Pancreatitis; BMI, body mass index; BUN, blood urea nitrogen; CRP, C-reactive protein; HCT, hematocrit; HR, heart rate; LDH, lactate dehydrogenase; RR, respiratory rate; SIRS, systemic inflammatory response syndrome; SOFA, Sequential Organ Failure Assessment; WBC, white blood cells.

**Table 2 pharmaceuticals-18-01601-t002:** Overview of studies assessing the rates and volumes of fluid therapy in acute pancreatitis.

ClinicalTrials.gov Identifier/Phase (If Specified)	Participants/Enrollment	Study Endpoint	Rate and Volume of Administration	Clinical Effects/Findings Results	Author, Year Reference
Retrospective cohort study	500 patients with MAP; Hydration group A (*n* = 252); hydration group B (*n* = 161); hydration group C (*n* = 87)	The first 12 h of vitals check	Hydration group A (0–1.5 mL/kg/h); Hydration group B (>1.5–3 mL/kg/h); Hydration group C (>3 mL/kg/h)	↓ opioid use ↓ readmission to the hospital	Doshi et al., 2020 [55]
Randomized controlled clinical trial	88 patients with AP; aggressive fluid arm (*n* = 43) or standard fluid arm (*n* = 45)	Development of SIRS, MODS; Hospital length of stay	Aggressive arm: 20 mL/kg bolus + 3 mL/kg/h for the first 24 h and then 30 mL/kg for the next 24 h Standard arm: 1.5 mL/kg/h for the first 24 h and 30 mL/kg during the next 24 h; 6400 mL vs. 2795 mL	↔ persistent SIRS ↔ pancreatic necrosis ↔ respiratory complications, AKI ↔ hospital stay length	Cuéllar-Monterrubio et al., 2020 [57]
Multicentre, open-label, parallel-group, randomized, controlled trial	249 patients with AP; aggressive fluid arm (*n* = 122) or moderate fluid arm (*n* = 127)	Development of MSAP/SAP during hospitalization; Fluid overload	Aggressive arm: 20 mL/kg bolus + 3 mL/kg/h Moderate arm: 10 mL/kg bolus + 1.5 mL/kg/h 5400 mL vs. 3310 mL	↑ fluid overload ↑ hospital stay	de-Madaria et al., 2022 [59]
Meta-analysis	Patients with AP; aggressive fluid arm (*n* = 1229) or moderate fluid arm (*n* = 1397)	NA	Aggressive: 3–5 mL/kg/h in first 24 h	↔ SIRS, MODS, pancreatic necrosis, mortality risk ↑ AKI, ARDS risk	Gad et al., 2020 [61]
Meta-analysis	3127 patients with SAP	NA	Aggressive: 250–500 mL/h or 5–10 mL/kg/h	↑ AKI, ARDS, persistent SIRS, pancreatic necrosis risk ↑ mortality risk ↑ hospital stay	Liao et al., 2022 [62]
Meta-analysis	4072 patients with AP; aggressive fluid resuscitation (*n* = 1653) or controlled fluid resuscitation (*n* = 2419)	NA	Aggressive: >250 mL/h or >5 mL/kg/h	↑ mortality ↑ MODS and infection risk ↑ hospital stay	Guo et al., 2023 [64]
Systematic review and meta-analysis	3423 patients with AP	NA		↑ AKI, ARDS risk ↔ mortality	Ding et al., 2023 [65]
Systematic review and meta-analysis	4667 patients with MAP, MSAP and SAP	NA	High: (≥20 mL/kg/h) Moderate: (≥10 to <20 mL/kg/h) Low: (5 to <10 mL/kg/h)	↑ clinical outcomes ↓ mortality	Kumari et al., 2024 [66]
Randomized controlled trial	44 patientd with AP; aggressive fluid arm (*n* = 22) or moderate fluid arm (*n* = 22)	Clinical improvement within 36 h; Decrease in HCT, BUN, creatinine and reduced epigastric pain level tolerance of oral intake, development of SIRS	Aggressive arm: 20 mL/kg bolus + 3 mL/kg/h Standard arm: 10 mL/kg bolus + 1.5 mL/kg/h 4886 mL vs. 3985 mL	↔ clinical improvement (expect of obese patients)	Angsubhakorn et al., 2021 [67]
Retrospective cohort study	10,400 patients with AP	30-day mortality; mechanical ventilation rates; severe sepsis	Aggressive: >3 mL/kg/h Standard: ≤1.5 mL/kg/h	↔ mortality ↔ hospital stay length	Tomanguillo et al., 2023 [70]
Retrospective cohort study	310 patients with AP	MODS; In-hospital mortality	Aggressive: ≥4.475 L/24 h Moderate: 2.8–4.475 L/24 h Conservative: <2.8 L/24 h	Aggressive fluid therapy increased MODS occurrence and hospital length stay	Messallam et al., 2021 [71]

Legend: ↑ activation or increase; ↓ inhibition or decrease; ↔ no impact. AKI, acute kidney injury; AP, acute pancreatitis; ARDS, acute respiratory distress syndrome; BUN, blood urea nitrogen; MODS, multiple organ dysfunction syndrome; MSAP, moderately severe acute pancreatitis; NA, not applicable; SAP, severe acute pancreatitis; SIRS, systemic inflammatory response syndrome.

**Table 3 pharmaceuticals-18-01601-t003:** Overview of selected studies investigating individual types of fluid in acute pancreatitis.

ClinicalTrials.gov Identifier/Phase (If Specified)	Participants/Enrollment	Type of Fluid	Rate of Administration	Clinical Effects/Findings	References
Double-blinded andomized controlled trial	331 patients with AP	RL solution, NS; Comparison of intravenous hydration with RL solution and NS	10 mL/kg bolus followed by continuous infusion at 3 mL/kg/h	RL: ↔ SIRS development ↓ hospital stay ↓ ICU admission ↓ local complications development	Lee et al., 2021 [90]
Prospective, single-center study, ranodmized controlled trial	142 patients with AP (77 with LR and 65 with NS)	RL solution, NS; Comparison of intravenous hydration with RL solution and NS	1000 mL within the first hour after randomization, and then 3 mL/kg/h until oral feeding	RL: ↓ AP severity ↓ CRP level	Kayhan et al., 2021 [97]
Open-label randomized controlled trial	51 patients with AP (26 with LR and 25 with NS)	RL solution, NS; Comparison of intravenous hydration with RL solution and NS	10 mL/kg within the first 60 min followed by infusion at the rate of 1.5 mL/kg/h until oral feeding	RL: ↓ SIRS development ↓ CRP level after 72 h	Karki et al., 2022 [98]
Systematic review and meta-analysis	549 patients with AP (230 with LR and 319 with NS)	RL solution, NS; Comparison of intravenous hydration with RL solution and NS	The regimen of fluid resuscitation differed across the included trials	RL: ↔ mortality ↔ SIRS development ↓ hospital stay	Aziz et al., 2021 [92]
Multicenter, stepped-wedge, cluster-randomized trial	259 patients with predicted SAP (112 with BMESs and 147 with NS)	BMESs, NS; Comparison of intravenous hydration with BMESs and NS	No detailed data	BMES: ↓ SIRS development ↓ organ failure	Ke et al., 2025 [95]
Meta-analysis	546 patients with AP	RL, NS; Comparison of intravenous hydration with RL and NS	The regimen of fluid resuscitation differed across the included trials	RL: ↔ mortality ↔ SIRS development ↓ ICU admission	Vedantam et al., 2022 [94]
Meta-analysis	248 patients with AP	RL, NS; Comparison of intravenous hydration with RL and NS	The regimen of fluid resuscitation differed across the included trials	RL: ↔ mortality ↔ SIRS, MODS, pancreatic necrosis development ↔ hospital stay ↓ ICU admission	Chen et al., 2022 [75]
Systematic review and meta-analysis	3752 patients with AP; aggressive fluid resuscitation (*n* = 1386) or controlled fluid resuscitation (*n* = 2366) for comparison intensify of fluid therapy; 631 patients with AP (251 with RL, 359 with NS and 21 with HES) for comparison type of fluid therapy	RL, NS, HES; Comparison of intravenous hydration with RL and NS, as well as aggressive and moderate fluid therapy	The regimen of fluid resuscitation differed across the included trials	RL: ↔ mortality ↓ hospital stay ↑ MODS risk	Dawson et al., 2023 [77]
Systematic review and meta-analysis	557 patients with AP (278 with RL and 279 with NS)	RL solution, NS; Comparison of intravenous hydration with RL solution and NS	The regimen of fluid resuscitation differed across the included trials	RL: ↓ mortality ↓ MODS, local complications risk ↓ hospital stay ↓ MSAP risk	Ocskay et al., 2023 [78]
Meta-analysis	431 patients with AP	RL solution, NS; Comparison of intravenous hydration with RL solution and NS	The regimen of fluid resuscitation differed across the included trials	RL: ↓ SIRS risk ↓ hospital stay ↓ local complications risk ↓ ICU admission	Wang et al., 2024 [79]
Meta-analysis	1424 patients with AP (651 with RL and 773 with NS)	RL solution, NS; Comparison of intravenous hydration with RL solution and NS	The regimen of fluid resuscitation differed across the included trials	RL: ↓ MSAP, SAP risk ↓ ICU admission ↓ local complications risk ↓ hospital stay	Hong et al., 2024 [80]
Retrospective analysis	20,049 admissions with AP	RL, NS; Comparison of intravenous hydration with RL and NS	The regimen of fluid resuscitation differed across the included trials	RL: ↓ 1-year mortality	Antoniak et al., 2023 [81]

Legend: ↑ activation or increase; ↓ inhibition or decrease; ↔ no impact. AP, acute pancreatitis; BMESs, balanced multielectrolyte solutions; HES, hydroxyethyl starch; ICU, intensive care unit; MODS, multiple organ failure; MSAP, moderately severe acute pancreatitis; NS, normal saline; RL, Ringer’s lactate; SAP, severe acute pancreatitis; SIRS, systemic inflammatory response syndrome.

## Data Availability

Data sharing is not applicable to this article as no datasets were generated or analyzed during the current study.

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
