# Peer review of "Fluid Therapy in Acute Pancreatitis—Current Knowledge and Future Perspectives"

_pharmaceuticals, 2025, doi:10.3390/ph18111601_

Round 1
Reviewer 1 Report
Comments and Suggestions for Authors
This review article provides a comprehensive and timely synthesis of the current evidence regarding fluid therapy in acute pancreatitis (AP). The topic is clinically relevant, well-structured, and supported by a thorough literature search. The authors effectively address key controversies and offer practical recommendations.
- This paper provides a thorough review of fluid therapy in acute pancreatitis. However, as it primarily adopts a narrative synthesis approach without conducting quantitative heterogeneity assessment or pooled analysis of effect sizes, the robustness of its conclusions may be somewhat weakened. It is recommended that the authors consider performing a meta-analysis if feasible. If not, the discussion should explicitly acknowledge the significant heterogeneity among studies and the absence of quantitative synthesis as a major limitation.
Author Response
Response to Reviewer 1 Comments
This review article provides a comprehensive and timely synthesis of the current evidence regarding fluid therapy in acute pancreatitis (AP). The topic is clinically relevant, well-structured, and supported by a thorough literature search. The authors effectively address key controversies and offer practical recommendations.
Response: We would like to thank the Reviewer for providing feedback and review. Thank you for this comment.
- This paper provides a thorough review of fluid therapy in acute pancreatitis. However, as it primarily adopts a narrative synthesis approach without conducting quantitative heterogeneity assessment or pooled analysis of effect sizes, the robustness of its conclusions may be somewhat weakened. It is recommended that the authors consider performing a meta-analysis if feasible. If not, the discussion should explicitly acknowledge the significant heterogeneity among studies and the absence of quantitative synthesis as a major limitation.
Response: Thank you for this comment. We agree with the above suggestion that the robustness of our conclusions may be somewhat weakened. Unfortunately, the prepared is a comprehensive, narrative review, and not meta-analysis (the performing meta-analysis was not feasible). Consequently, according to your suggestions, we provided information about indicated limitations in the last section (8. Conclusions).

Reviewer 2 Report
Comments and Suggestions for Authors
Review of the manuscript entitled “Fluid therapy in acute pancreatitis- current knowledge and future perspectives”.
The article deals with a very important topic, which is fluid therapy in acute pancreatitis (AP). The approach to fluid therapy has changed several times over the last decade, and it is still difficult to achieve a consensus that could provide a basis for formulating certain guidelines.
The publication is very extensive, the authors approached the subject reliably, discussing individual aspects of the topic in great detail.
In the introduction, the authors present the pathogenesis of acute pancreatitis and doubts regarding the most beneficial type of fluids as well as the rate at which ones should be administered. In chapter “The rationale for fluid therapy in AP” the authors present the pathogenesis leading to fluid and electrolyte disturbances in AP. The chapter on AP severity includes numerous scales and their characteristics.
In the chapter on fluid therapy in AP, the authors extensively discuss the history of fluid therapy in AP, ranging from more to less aggressive management. This analysis is based on current, carefully selected literature. The authors discuss the volume of fluids, the rate, and the type of fluid.
Two tables are particularly valuable:
-table 2: Overview of studies assessing the rates and volumes of fluid therapy in AP and
-table 3: Overview of studies investigating individual types of fluid in AP.
A brief summary highlights the importance of individualizing fluid therapy for AP, with consideration given to less aggressive management. Two studies currently in progress may provide evidence to support this approach.
I have no significant, substantive comments to the publication.
I recommend this article for publication in “Pharmaceuticals”.
Author Response
Response to Reviewer 2 Comments
Review of the manuscript entitled “Fluid therapy in acute pancreatitis- current knowledge and future perspectives”.
The article deals with a very important topic, which is fluid therapy in acute pancreatitis (AP). The approach to fluid therapy has changed several times over the last decade, and it is still difficult to achieve a consensus that could provide a basis for formulating certain guidelines.
The publication is very extensive, the authors approached the subject reliably, discussing individual aspects of the topic in great detail.
In the introduction, the authors present the pathogenesis of acute pancreatitis and doubts regarding the most beneficial type of fluids as well as the rate at which ones should be administered. In chapter “The rationale for fluid therapy in AP” the authors present the pathogenesis leading to fluid and electrolyte disturbances in AP. The chapter on AP severity includes numerous scales and their characteristics.
In the chapter on fluid therapy in AP, the authors extensively discuss the history of fluid therapy in AP, ranging from more to less aggressive management. This analysis is based on current, carefully selected literature. The authors discuss the volume of fluids, the rate, and the type of fluid.
Two tables are particularly valuable:
-table 2: Overview of studies assessing the rates and volumes of fluid therapy in AP and
-table 3: Overview of studies investigating individual types of fluid in AP.
A brief summary highlights the importance of individualizing fluid therapy for AP, with consideration given to less aggressive management. Two studies currently in progress may provide evidence to support this approach.
I have no significant, substantive comments to the publication.
I recommend this article for publication in “Pharmaceuticals”.
Response: We would like to thank the Reviewer for providing feedback and review. Thank you for above comments.

Reviewer 3 Report
Comments and Suggestions for Authors
The authors clearly explained the concept of Fluid therapy in acute pancreatitis: current Knowledge and future perspectives. Explained the adverse effects and clinical trials data very well.
Author Response
Response to Reviewer 3 Comments
The authors clearly explained the concept of Fluid therapy in acute pancreatitis: current Knowledge and future perspectives. Explained the adverse effects and clinical trials data very well.
Response: We would like to thank the Reviewer for providing feedback and review. Thank you for above comments.

Reviewer 4 Report
Comments and Suggestions for Authors
Dear Authors,
I congratulate for this comprehensive study on fluid treatment in AP. In this study, research related to fluid therapy in the AP over the past 5 years has been examined in detailed and summarized. I have some concerns about the manuscript.
Concern 1. I think the main text is too long. You can make the introduction and severity of AP sections more concise.
Concern 2. Since many of the meta-analyses you included in Table 2 and Table 3 used the same studies, they produced similar results. For example, the publications in references 74, 75, 76, and 93 evaluated the same studies. I believe it would be more appropriate to select the more comprehensive ones from these references and exclude the others from the tables.
Sincerely
Author Response
Response to Reviewer 4 Comments
Dear Authors,
I congratulate for this comprehensive study on fluid treatment in AP. In this study, research related to fluid therapy in the AP over the past 5 years has been examined in detailed and summarized. I have some concerns about the manuscript
Response: We would like to thank the Reviewer for providing feedback and review. Thank you for this comment.
Concern 1. I think the main text is too long. You can make the introduction and severity of AP sections more concise.
Response: Thank you for this comment. We agree with the above suggestion. That is why we shortened the Introduction section and the Severity of AP section. In our opinion, the remaining information and data in these section are very valuable and necessary to present and understand fully issue of fluid therapy in AP.
Concern 2. Since many of the meta-analyses you included in Table 2 and Table 3 used the same studies, they produced similar results. For example, the publications in references 74, 75, 76, and 93 evaluated the same studies. I believe it would be more appropriate to select the more comprehensive ones from these references and exclude the others from the tables.
Response: Thank you for this comment. We agree with the above suggestion. That is why we removed from Table 2 and Table 3 publications with smaller study groups and presenting similar results, and left meta-analysis with big study groups.
